# A spatial gradient of bacterial diversity in the human oral cavity shaped by salivary flow

Diana M. Proctor ⓘ [1,2,3], Julia A. Fukuyama[4], Peter M. Loomer[3,5], Gary C. Armitage[3], Stacey A. Lee[3], Nicole M. Davis[1], Mark I. Ryder[3], Susan P. Holmes ⓘ [6] & David A. Relman ⓘ [1,2,7]

Spatial and temporal patterns in microbial communities provide insights into the forces that shape them, their functions and roles in health and disease. Here, we used spatial and ecological statistics to analyze the role that saliva plays in structuring bacterial communities of the human mouth using >9000 dental and mucosal samples. We show that regardless of tissue type (teeth, alveolar mucosa, keratinized gingiva, or buccal mucosa), surface-associated bacterial communities vary along an ecological gradient from the front to the back of the mouth, and that on exposed tooth surfaces, the gradient is pronounced on lingual compared to buccal surfaces. Furthermore, our data suggest that this gradient is attenuated in individuals with low salivary flow due to Sjögren's syndrome. Taken together, our findings imply that salivary flow influences the spatial organization of microbial communities and that biogeographical patterns may be useful for understanding host physiological processes and for predicting disease.

[1] Department of Microbiology and Immunology, Stanford University School of Medicine, Stanford, CA 94305, USA. [2] Infectious Diseases Section, Veterans Affairs Palo Alto Health Care System, Palo Alto, CA 94304, USA. [3] Division of Periodontology, University of California, San Francisco School of Dentistry, San Francisco, CA 94143, USA. [4] Department of Computational Biology, Fred Hutchinson Cancer Research Institute, Seattle, WA 98109, USA. [5] Ashman's Department of Periodontology and Implant Dentistry, New York University College of Dentistry, New York, NY 10010, USA. [6] Department of Statistics, Stanford University, Stanford, CA 94305, USA. [7] Department of Medicine, Stanford University School of Medicine, Stanford, CA 94305, USA. Correspondence and requests for materials should be addressed to D.A.R. (email: relman@stanford.edu)

dentifying spatial patterns of variation in microbial community composition is necessary for understanding the mechanisms that give rise to patterning and the factors that maintain or disrupt it. In landscape ecology, communities are known to follow a set of characteristic spatial patterns—exhibiting homogenous, random and patchy, or gradual distributions across space[1,2]. Currently, few studies of host-associated microbiota have explored the types and extent of spatial patterns characteristic of community variation within gross anatomic sites[3–7], particularly at spatial scales that encompass an entire anatomic region.

Cavities and other forms of dental disease, such as chronic periodontitis, demonstrate a remarkable degree of site specificity[8,9]. This is one reason that the human oral cavity is an excellent body site for examining mechanisms underlying spatial patterns in the human microbiota. Other reasons include the feasibility of detailed spatial sampling, the wealth of unique microbial habitats such as, soft vs. hard tissues and keratinized vs. non-keratinized soft tissues, and aspects of host physiology that create environmental gradients. For example, proximity to the nearest major salivary gland determines the velocity of the salivary film flowing over individual surfaces[10]; flow velocity in turn determines the rate that molecules such as sucrose and acids are cleared from different oral compartments[11,12]. In healthy individuals, salivary film velocity varies considerably between the front and the back of the mouth, and oral clearance is known to be faster from lingual sites compared to buccal ones. As a consequence, salivary flow is a major determinant of microbial metabolic potential and intra-plaque pH[13,14], both of which likely vary according to salivary film velocity, and thus tooth position, even in healthy individuals.

Several experimental systems demonstrate that the loss of salivary flow results in a site-specific shift in the spatial pattern of dental cavities. In healthy individuals with normal salivary flow, caries tend to be restricted to the biting and inter-proximal surfaces of teeth particularly impacting the molars and pre-molars[8,15]. An experimental model in which salivary glands were surgically removed from rats revealed mesial, distal and lingual smooth surfaces, particularly sites in the lower jaw, to be especially susceptible to caries following the onset of hyposalivation[16]. In humans, several disparate patient populations experience low salivary flow and similarly exhibit an increased burden, compared to controls, of root, cervical, and smooth surface caries of the incisors and canines[17–19]. Although aging is arguably an independent predictor of low salivary flow[18,20–22], a definite endogenous and systemic cause is the progressive inflammatory autoimmune disorder Sjögren's syndrome (SS) that results in the gradual loss of salivary and lacrimal gland function[23]. In addition, iatrogenic causes of low salivary flow include the use of any one of >400 different medications spanning virtually every medication class[24] as well as radiation therapy to the head or neck[25].

The aberrant metabolic activity of the microbiota largely drives cariogenesis[26]. Therefore, this characteristic shift in the spatial pattern of caries, from the back to the front of the mouth, during states of hyposalivation implies that salivary flow typically plays a role in shaping the spatial organization of microbial communities in human health. Yet despite longstanding knowledge of this relationship[13], our understanding of how salivary flow impacts the spatial patterning of the microbial consortia across teeth remains limited. This deficit in our knowledge is likely related to the widespread use of sample types and collection methods that are ill-suited for studying biogeographical position effects[27–30]. Sample types such as saliva, and methods such as rinsing samples, or pooled dental plaque samples, which average communities from a wide array of intra-oral habitats, cannot be used to assess fine-scale spatial effects.

In this study, we sought to identify the type and extent of spatial patterns formed by bacterial communities inhabiting the oral cavity before investigating, as part of an ongoing larger study, the impact of low salivary flow on observed spatial patterns. Our analysis suggests that bacterial communities inhabiting the molars and incisors, of healthy humans, can be distinguished from one another. In addition, our data indicate that communities inhabiting soft and hard tissues vary across the anterior to posterior dimension of the mouth in a manner consistent with an ecological gradient, despite the profound differences between these intra-oral habitats. Finally, we provide evidence that the anterior–posterior gradient is modulated in patients with low salivary flow due to SS. Taken together, these results imply that salivary flow plays a role in structuring the community gradient. The reproducible framework used here to identify specific spatial patterns in the variation of human indigenous communities (and associated processes) may be extended to other body sites, improving our understanding of how spatial patterns and processes contribute to human health and disease.

## Results

**Overview of data and patient cohorts.** Bacterial community taxonomic count data (3 data sets) were generated from a total of 9449 samples collected from 31 individuals (Supplementary Table 1). A discovery data set comprised 16S ribosomal RNA (rRNA) gene (V4–V5 region) amplicon sequences from 1905 supragingival plaque samples (median sequencing depth per sample, 2258) surveying the buccal and lingual aspects of the first molars and central incisors of 11 individuals. A validation data set comprised V4 amplicon sequences from 7002 supragingival plaque samples (median sequencing depth, 73,772) of the buccal and lingual aspects of all teeth (excluding third molars) in 19 additional individuals, including 9 healthy controls and 10 individuals with low salivary flow due to SS. Finally, a mucosal biogeography data set comprised V4 amplicon sequences from 168 additional samples of supragingival plaque and 374 samples of the buccal mucosa (BM), alveolar mucosa (AM), and keratinized gingiva (KG) (median sequencing depth, 82,354) in 3 additional healthy individuals.

**Overview of bacterial taxonomic representation.** After filtering 16S rRNA gene sequences to exclude contaminants (Supplementary Methods; Supplementary Data 1), the largest (validation) data set revealed 480 unique bacterial amplicon sequence variants (ASVs), which represent highly resolved (e.g., strain-level) taxonomic units[31,32]. Overall, representatives of 13 bacterial phyla were detected in the validation data set. Consistent with published findings[33,34], the five most abundant phyla (Firmicutes, Proteobacteria, Actinobacteria, Bacteroidetes, and Fusobacteria) accounted for 99.9% of all reads (Supplementary Table 2). A total of 118 unique genera were identified, and the 10 most abundant genera accounted for 88% of all reads (Supplementary Table 3; *Streptococcus*, *Haemophilus*, *Rothia*, *Neisseria*, *Actinomyces*, *Corynebacterium*, *Veillonella*, *Abiotrophia*, *Gemella*, and *Prevotella*).

**Communities on the molars and incisors are distinct.** The composition of oral bacterial communities varied by subject (Fig. 1a), tooth class (Fig. 1b), and tooth aspect (Fig. 1c, d) as shown in the principal coordinates analysis (PCoA) on Bray Curtis dissimilarity. The first coordinate explained 23.48% of the total variation and partially separated molar and incisor communities: 69% of incisor samples mapped to positive axis 1 scores of which about half exceeded 0.25 while only 26% of molar samples mapped to positive axis 1 scores with fewer than 0.7% exceeding 0.25 (Fig. 1b). Strikingly, 54% of all lingual

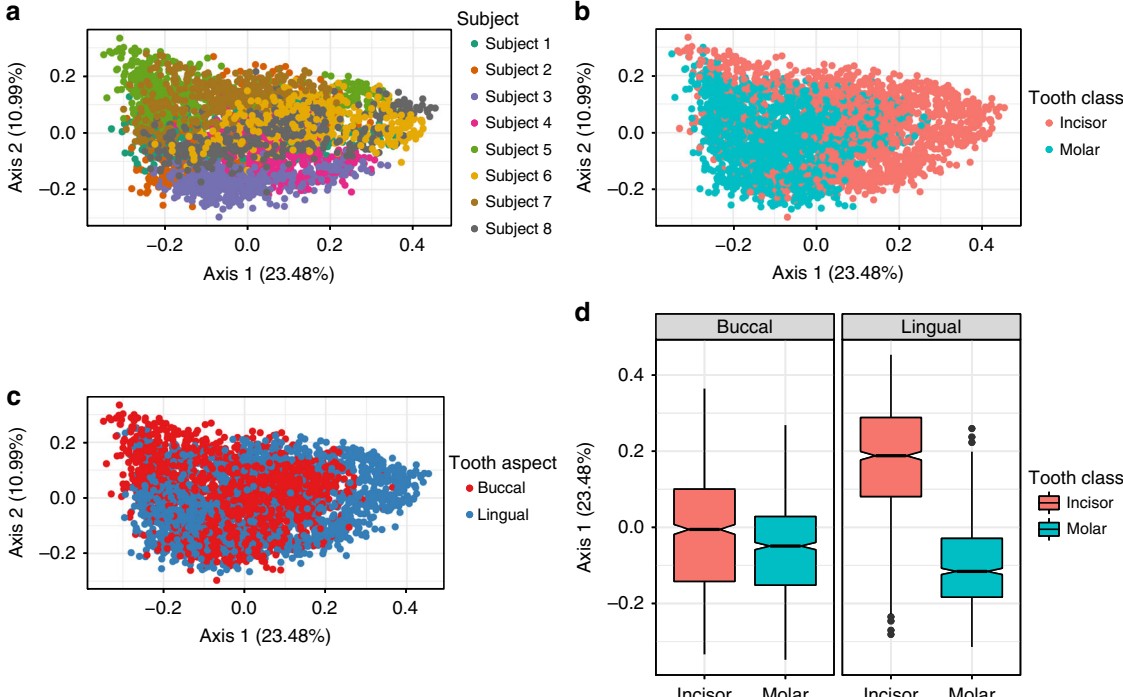

**Fig. 1** Subject, tooth class, and tooth aspect explain variation in oral communities. Adonis and PCoA on Bray Curtis dissimilarity (90 taxa, 3393 samples) revealed separation of communities by **a** subject, **b** tooth class, and **c** tooth aspect. **d** Each boxplot box demarcates the first and third quartiles of the PCoA axis 1 scores, while the horizontal black lines within each box define the median PCoA axis 1 scores. Boxes are colored according to tooth class (incisor, molar). Boxplots comparing buccal molar communities ($N = 849$) to those on the buccal incisors ($N = 849$), lingual molars ($N = 846$), and lingual incisors ($N = 849$) revealed an interaction between tooth class and tooth aspect, highlighting the greater separation of molar and incisor communities at the lingual, as compared to the buccal, tooth aspect

samples also mapped to positive axis 1 (Fig. 1c), which suggests an interaction between tooth class and tooth aspect. Indeed, axis 1 scores separate molar and incisor communities from one another only if samples originated from the lingual, but not the buccal, side of teeth (Fig. 1d). Exploration of the second, third, and fourth principal coordinates suggested that axis 1 best captured the difference between molars and incisors (Supplementary Figure 1; Supplementary Note 1).

Since the PCoA revealed significant overlap in community composition across the variables, subject, tooth class, and tooth aspect, we sought to quantify the fraction of variance explained by each factor using an analysis of dissimilarity (Adonis). After accounting for the non-independence of temporal replicates, interpersonal variation explained 46.9% of the total variance while the difference between molars and incisors (tooth class) accounted for 9.4% and tooth aspect (buccal vs. lingual) accounted for 3.8% (Supplementary Note 1). The interaction between subject and tooth class accounted for an additional 9.5% of the variation, suggesting that physiological differences or behavioral habits (or both) between subjects contributed to the variability observed between the molar and incisor communities. The interaction between tooth class and tooth aspect accounted for 4.3% of total variation indicating that the communities on different aspects (buccal, lingual) of individual teeth represent unique community patches. These patterns in community composition were robust to a variety of data transformation methods (Supplementary Figure 2; Supplementary Note 1) and distance metrics (Supplementary Figure 3; Supplementary Note 1) and were consistent with findings from the discovery data set (Supplementary Data 2), drawn from observations on 11 different individuals, samples of which were sequenced on a different platform to a lower sequencing depth.

The rate of decay in community similarity over time was modest for all subjects (Supplementary Figure 4; Supplementary Note 2), confirming previous reports[35] that oral microbial consortia in the aggregate tend to be relatively stable despite fluctuations in the abundances of individual ASVs over time. Molar communities were significantly more diverse (Wilcoxon rank sum: Shannon ($p < 0.05$), Simpson ($p < 0.05$), Chao1 ($p < 0.05$)) than incisor communities, differences that were not associated with sequencing depth (Wilcoxon rank sum, $p > 0.1$). Taken together, these data suggest that the communities inhabiting the molars and incisors differ, particularly on the lingual surfaces of teeth.

**Supragingival communities segregate along a gradient**. Since the observed difference between molars and incisors suggested non-random patterns of spatial variation in community composition, we next determined the type of spatial pattern that best fit the data, specifically that of an ecological gradient or habitat fragmentation and patchiness associated with tooth classes. We collected samples of the buccal and lingual aspects of all teeth (excluding third molars) from each of 7 individuals and performed a trend surface analysis (117 taxa; 1701 samples), a method used to analyze multivariate spatial patterns in community ecology.

Community composition varied across tooth class in a manner suggestive of an ecological gradient. Regardless of tooth aspect (buccal, lingual), communities on the central incisors (teeth 8–9, 24–25) were associated with lower axis 1 scores compared to communities on the molars (teeth 2–3, 14–15, 18–19, 30–31) (Fig. 2a). Axis 1 scores for communities on the remaining tooth classes (lateral incisors, canines, and pre-molars) gradually

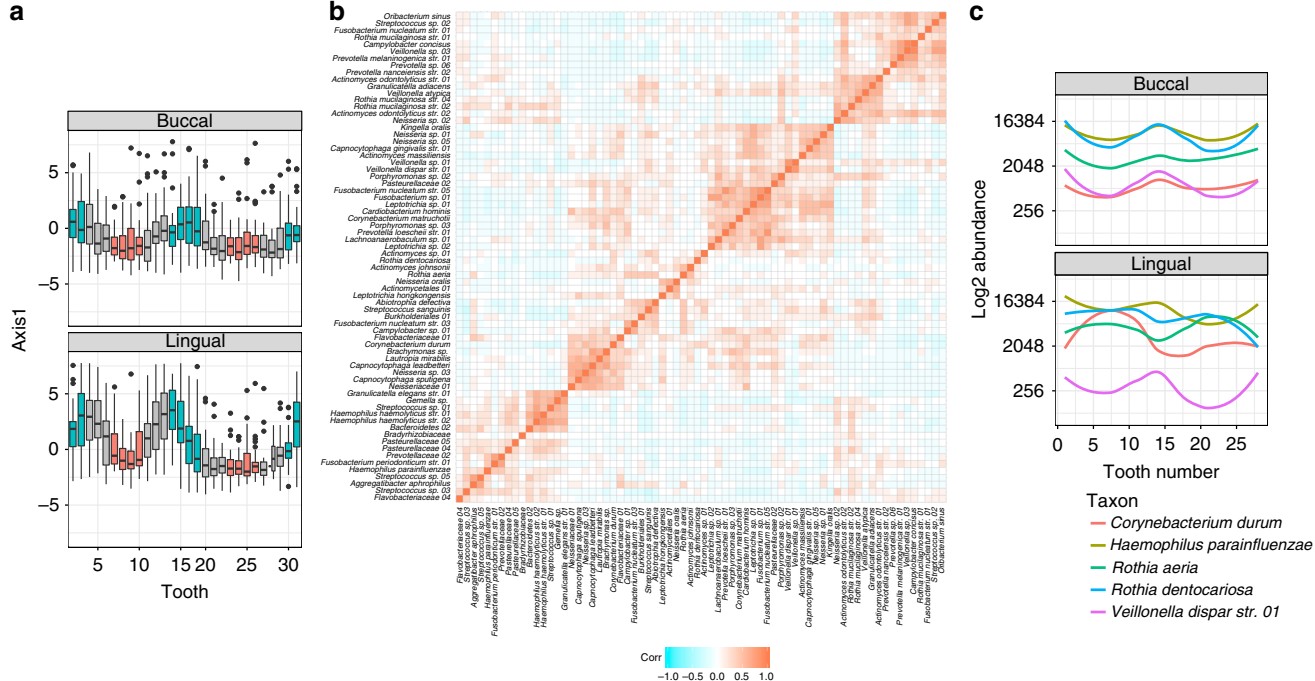

**Fig. 2** Supragingival communities vary by tooth class in a structured fashion suggestive of a gradient. **a** Trend surface analysis (TSA) was performed on the species by sample data matrix (117 taxa, 1701 samples). TSA axis 1 scores (y axis) are plotted as a function of universal tooth number (x axis). Each box demarcates the first and third quartiles of the TSA axis 1 scores, for each tooth, while the horizontal black lines within each box define the median TSA axis 1 scores. Boxes are colored according to tooth class (molars, incisors, other tooth classes) and samples from the lingual and buccal tooth aspects are shown separately. Molar communities (lingual $N = 248$; buccal $N = 249$) appeared to be associated with more positive scores along axis 1 compared to incisor communities (lingual $N = 250$, buccal $N = 249$) that shared more negative axis 1 scores. Of special interest, canine (lingual $N = 125$; buccal $N = 126$) and premolar (lingual $N = 227$; buccal $N = 227$) communities appeared to be arrayed in an ordered fashion in between the molar and incisor poles. **b** The between-taxa correlation matrix arranged by hierarchical clustering with average linkage revealed groups of taxa that shared positive or negative correlations in their distributions across samples and **c** exhibited a significant pattern of spatial dependence in their distributions across sites (Moran's $I$, adj. $p < 0.05$)

increased in a stepwise, ordered fashion from lower toward higher axis 1 scores (Fig. 2a). In other words, the central incisors and first molars corresponded to opposing poles of a gradient in community composition.

Interestingly, the top 10 most abundant taxa exhibited a relatively homogenous distribution across space: when the analysis was constrained to just these taxa, the community gradient could not be detected for either jaw or tooth aspect (Supplementary Figure 5). Rather, the gradient emerged for all sites only after 20 taxa were analyzed, and it became increasingly pronounced as additional taxa were incorporated into the analysis (Supplementary Figure 5).

Since the trend surface analysis is a decomposition of the correlations between taxa, we next examined a matrix of between-taxa correlations explicitly through hierarchical clustering (Fig. 2b). Notable patterns emerged when comparing the distributions of taxa within distinct clusters, across teeth. For example, *Veillonella dispar* str. 01 and *Haemophilus parainfluenzae* appeared to be enriched on the molars compared to the incisors regardless of tooth aspect or jaw (Fig. 2c). In contrast, *Corynebacterium durum*, *Rothia dentocariosa*, and *R. aeria* were enriched on the molars compared to the incisors only on the buccal surfaces of teeth; in contrast, on lingual surfaces, these organisms were enriched on the incisors compared to molars and other sites. Of these, all but *R. dentocariosa* were found to vary significantly as a function of the distance separating sites (Moran's *I*, adj. $p < 0.05$). In fact, of the top 70 taxa, 52 varied significantly as function of the geometric distance separating sites (Supplementary Figure 6).

Analysis of the other significant ordination axes revealed not only the differentiation of communities along the anterior–posterior dimension but also separation of communities based on tooth aspect and jaw (Supplementary Figure 7; Supplementary Note 3). The robustness of these findings was assessed by comparison to a principal coordinates of neighbor matrices (PCNM; Supplementary Figure 8; Supplementary Note 4) and model selection of 20 Moran's eigenvector maps (MEM; Supplementary Figure 9; Supplementary Note 5). All three analyses (trend surface, PCNM, MEM) independently identified the anterior–posterior gradient as a significant spatial structure. Taken together, these results suggest that the gradient in community composition between the anterior and posterior mouth is unlikely an artifact of method, but instead reflects underlying differences in the abundance profiles of spatially variant taxa across teeth.

**Mucosal communities conform to anterior–posterior gradient.** We hypothesized that the community gradient might reflect an underlying large-scale environmental variable shaped by salivary flow. To evaluate whether the observed spatial pattern was determined by a large-scale gradient or the morphological features of teeth (e.g., tooth size, tooth shape, tooth surface, and/ or tooth age—as measured since time of permanent tooth eruption), we recruited three additional healthy individuals and performed a trend surface analysis on samples from the supragingival surfaces of their teeth as well as the BM, AM, and KG adjacent to each tooth and tooth aspect.

Communities inhabiting the buccal and lingual aspects of all sites, including soft tissues, conformed to an ecological gradient that distinguished communities inhabiting the front of the mouth from those inhabiting the back (Fig. 3). The smile-shaped curves reflected continuous and gradual variation in community composition across the anterior–posterior dimension. Communities on or near the incisors (#7–10, 23–26) tended to share negative axis 1 scores; those on or near the molars (#2–3, 14–15, 18–19, 30–31) shared positive scores; and those near or on remaining tooth classes were arrayed in an ordered fashion between the incisor and molar poles. Interpretation of other axes highlighted the anterior to posterior gradient as well as differences between jaws (Supplementary Figure 10; Supplementary Note 6).

The gradient was detected for shedding and non-shedding surfaces alike despite the clear differentiation of communities at these habitats in a PCoA on Bray Curtis dissimilarity (Supplementary Figure 11). These data suggest that factors associated with tooth morphology alone cannot explain the gradient in community composition and that the spatial extent of the anterior–posterior gradient encompasses the entire oral cavity.

**Salivary flow affects oral microbial community composition.** Next we examined several clinical variables, including salivary flow, that distinguished between control samples and samples from individuals with generally low salivary flow rates (Supplementary Note 7; Supplementary Figure 12) and the impact of these variables on the composition of tooth-associated bacterial communities. Communities clustered in a constrained correspondence analysis by the health status (SS or control) of the human host (Fig. 4a). Strikingly, communities from the one SS subject who had a high unstimulated whole-salivary flow rate (UWS-FR) grouped with communities from the healthy controls (Fig. 4b, e). Similarly, communities from the SS subjects who had higher stimulated whole-salivary flow rates (SWS-FRs) grouped with controls rather than with low-flow-associated communities (Fig. 4c, f). On the other hand, communities did not appear to

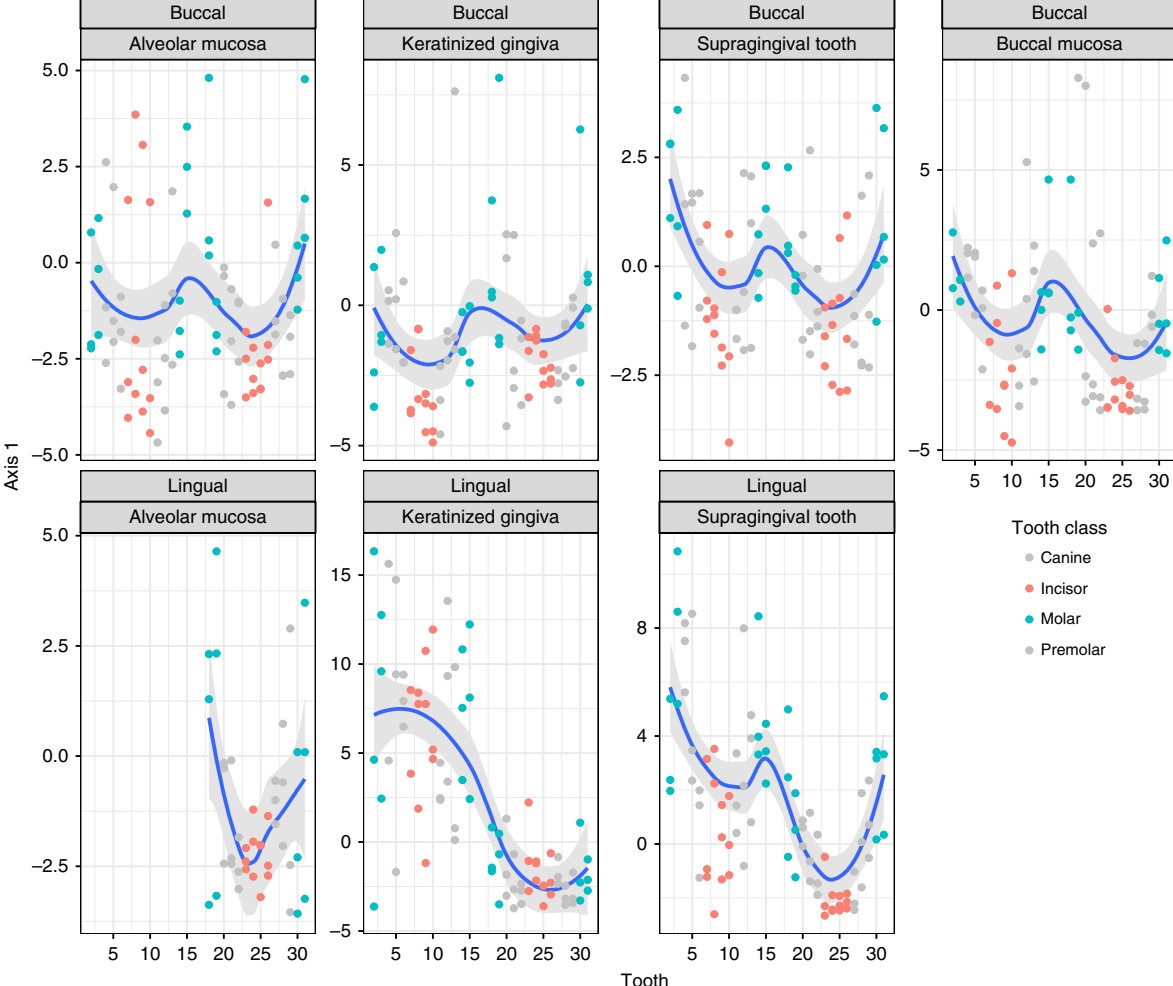

**Fig. 3** Mucosal communities also vary along an anterior to posterior gradient. A trend surface analysis (TSA) on samples of the oral mucosa and supragingival surfaces of 3 additional individuals (101 taxa, 542 samples) captured 10.5% of the total variation in the data, with the first axis accounting for 49% of the explained variance. TSA Axis 1 scores (y axis) are plotted as a function of tooth number (x axis). Each panel represents a unique tooth aspect (buccal, lingual) and intra-oral habitat (alveolar mucosa, keratinized gingiva, supragingival plaque, or buccal mucosa). Each point corresponds to a community sample, and samples are colored according to tooth class (molar, incisor, other tooth class). Blue curves denote local regression (loess) fits with associated 95% confidence intervals shaded in gray. Molar communities tended toward more positive scores along axis 1 compared to incisor communities, which tended toward more negative scores along axis 1 compared to molars. The remaining tooth classes were structured in an ordered fashion between the molar and incisor poles. These data suggest that both supragingival and soft tissue communities vary along a gradient from the front to the back of the mouth

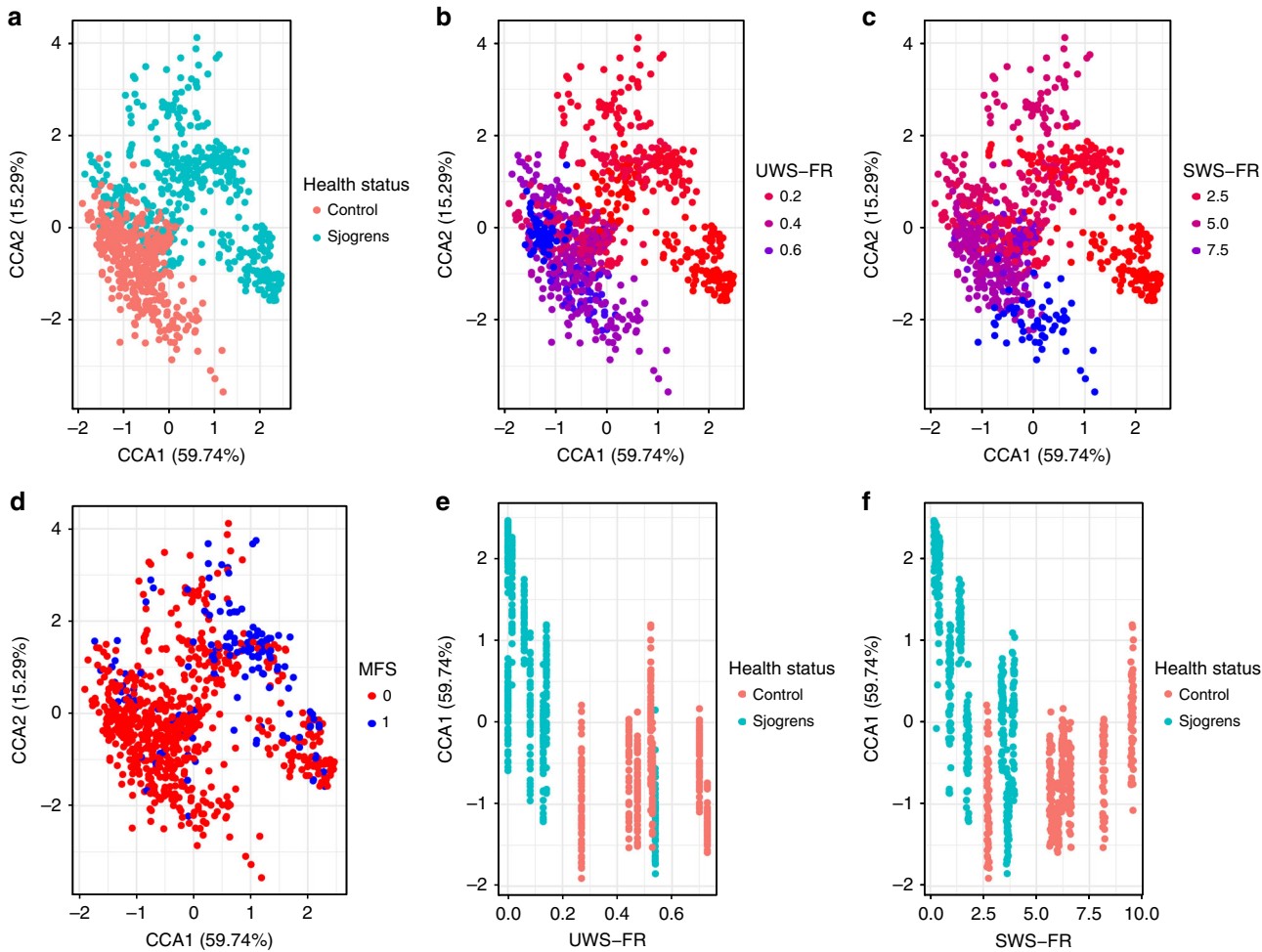

**Fig. 4** Low salivary flow impacts the composition of microbial communities independent of past dental caries. Constrained correspondence analysis (CCA) was performed on the species × sample data matrix for the validation data set (147 taxa; 825 samples). Each point corresponds to a community sample colored by **a** health status (Sjögren's, Control), **b** unstimulated whole-salivary flow rate (UWS-FR: 0.2, 0.4, or 0.6 mL/min), **c** stimulated whole-salivary flow rate (SWS-FR: 2.5, 5.0, or 7.5 ml/min), or **d** the index of missing, filled surfaces for smooth surface sites (MFS: 0, 1). The clustering of samples suggested that UWS-FR **b** and SWS-FR **c** explain more of the variation in the data than health status **a** or MFS **d**. To examine this explicitly, **e** CCA1 scores were plotted against UWS-FR, and community samples were colored according to health status of the host (Sjögren's, Control). Samples from the one Sjögren's patient who had a relatively high UWS-FR (~0.5 ml/min) grouped along CCA1 with control samples rather than with the other Sjögren's samples. Likewise, **f** samples from the healthy control patient who had a relatively low SWS-FR (~2.5 ml/min) grouped along CCA1 with Sjögren's samples rather than with other control samples

cluster strongly with the missing, filled surfaces (MFS) index (Fig. 4d), as might be expected if caries were the dominant factor influencing community composition. Indeed, a permutation test revealed that UWS-FR ($F = 115.32$, $p < 0.01$) and SWS-FR ($F = 27.76$, $p < 0.01$) but not MFS ($F = 8.9$, $p = 0.84$) contributed to the variation that distinguished communities collected from these subject cohorts.

The distribution of taxa along axis 1 revealed negative scores (where 89% of control samples mapped) for a variety of health-associated taxa (Supplementary Figure 13), including *Abiotrophia defectiva*, two *Capnocytophaga* spp., *Fusobacterium* sp., *Lautrophia mirabilis*, three *Leptotrichia* spp., two *Rothia* spp., and *Streptococcus sanguinis*[7]. On the other hand, positive scores (where 67% of Sjögren's samples mapped) were associated with a variety of acid-loving/producing or caries-associated taxa such as *Catonella sp.*, *S. mutans*, *Lactobacillus fermentum*, *Scardovia wiggsiae*, two *Atopobium parvulum* strains, and *Veillonella* spp., among others[36,37]. Taxa associated with poor oral health status such as *Megasphaera* sp. and *Oribacterium* spp. were also associated with positive axis 1 scores[38].

Taken together, these data suggest that low salivary flow modulates community composition, selecting for acid-loving and acid-producing species, possibly as a result of the homogenization of intra-plaque pH, which in healthy humans varies across sites in the oral cavity[13].

**Salivary flow impacts the organization of oral microbiota.** Next, we examined the impact of low salivary flow on the spatial organization of bacterial communities (Fig. 5). In 3 of the 10 Sjögren's patients (Sjögren's 04, 05, 06), the gradient appeared to be completely attenuated—in these individuals, the distribution of communities across sites appeared invariant with incisor and molar communities sharing similar Axis 1 scores in both jaws. In four other Sjögren's patients (Sjögren's 07, 08, 09, 10), the gradient was partially altered or attenuated in one or both jaws, whereas the distribution of communities for the three remaining SS subjects (Sjögren's 01, 02, 03) could be described as conforming to an anterior–posterior gradient similar to controls. For each of the control subjects with normal salivary flow rates, the

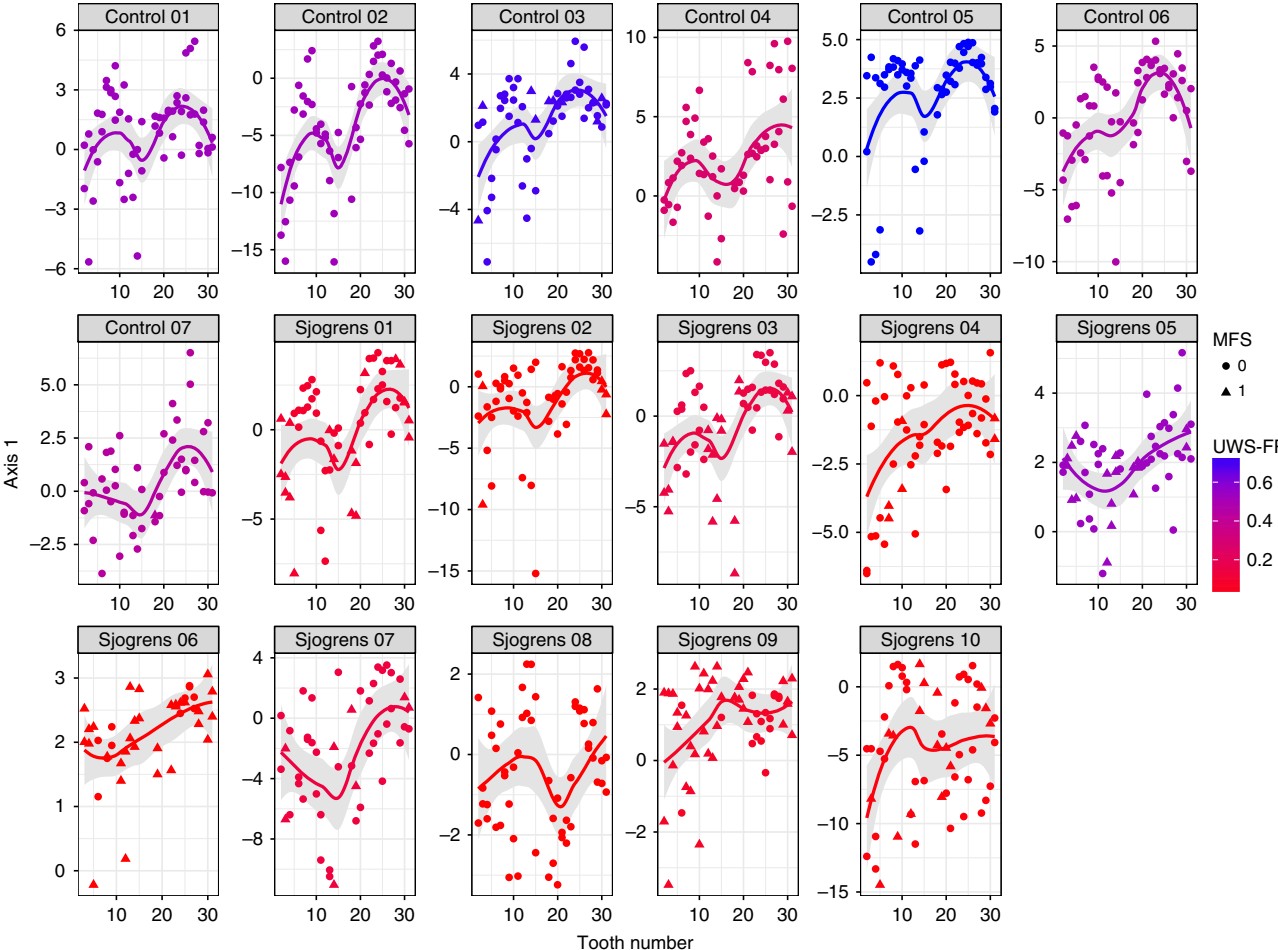

**Fig. 5** Low salivary flow is associated with gradient modulation. A trend surface analysis was performed on supragingival samples collected from 10 low-flow subjects with Sjögren's and 7 healthy controls. Each panel encompasses samples collected from an individual subject (control 1–7; Sjögren's 1–10). The color of the local (loess) regression lines corresponds to the UWS-FR (low flow, moderate flow, high flow) for that subject while the gray shading encompasses the 95% confidence interval surrounding each model fit. Each point is a community sample, and shapes map to the presence (1) or absence (0) of past caries experience at each smooth surface. The gradient appears to be attenuated or modulated in most of the low-flow cohort (Sjögren's) compared to otherwise healthy individuals (controls)

distribution of communities across sites appeared to be consistent with the previously characterized community gradient. In fact, molars and incisors were only significantly different along axis 1 from each other across all quadrants and tooth aspects for the control subjects and for Sjögren's subjects 01–03 (Wilcoxon rank sum, $p < 0.05$; Supplementary Data 3). Since the gradient was altered or attenuated in most low-flow subjects, as well as in one jaw for a control individual with a relatively low flow rate, these data suggest that salivary flow may play a role in establishing or maintaining the gradient. Elements of metacommunity structure (EMS) analysis provided further insight into the differences between communities in the low flow and control cohorts (Supplementary Figure 14; Supplementary Note 8).

## Discussion

In this study, we used spatial and ecological statistics to understand spatial patterning in the human oral cavity. By focusing on shared rather than differential patterns of community variation, we found that communities vary geographically within several distinctive habitats—the teeth, BM, AM, and KG—in a manner consistent with an anterior-to-posterior gradient. We present

preliminary data from patients with low saliva production, supporting a role for salivary flow in structuring the gradient.

Our work shows that communities inhabiting soft and hard intra-oral tissues alike conform to an anterior–posterior gradient in community composition. Our data are consistent with previous work identifying soft and hard tissues as distinct microbial habitats[33,39–41], which makes the shared spatial pattern identified here particularly striking. Prior evidence supports our finding that the microbial consortia vary between the front and back of the mouth though extant studies have not identified the spatial pattern of variance as a gradient and have only examined tooth-associated surfaces[42–45]. Our data likewise support prior observations that the difference between molar and incisor communities is unlikely an artifact of differences between sites in plaque biomass[44] although biomass undoubtedly influences the composition of communities[46]. Taken together, these data suggest that the gradient is a reflection of a large-scale environmental factor that selects for a gradual change in the abundances of a large fraction of non-core taxa across sites. While the 10 most abundant "core" taxa did not exhibit site specificity, the vast majority of the others did, indicating that the most abundant taxa may be habitat generalists, whereas lesser abundant taxa exhibit site

specificity. While one research group used patterns of co-occurrence to identify microbial interactions as a likely explanation for the differentiation of molar and incisor communities[43], other non-mutually exclusive explanations include habitat selection and historical contingencies[47].

Salivary flow appears to play a role in maintaining the gradient in community composition in healthy individuals. Other less likely explanations for the observed gradient include abrasion related to the mechanical movement of the tongue, oral hygiene practices, and the known temperature differential between the front and back of the mouth[48,49]. Though these factors were not evaluated explicitly in this study, they cannot explain the gradient modification in individuals with low salivary flow or the existence of a gradient on all three mucosal surfaces, nor would they be expected to select for caries-associated bacteria as reported here.

In healthy humans, salivary film velocity and oral clearance rates vary between the front and the back of the mouth[10,11]. As a consequence, in healthy humans, the duration of plaque exposure to dietary and microbial-derived acids, and hence pH, varies across the anterior to posterior dimension—and, a reduction in salivary flow results in prolonged periods of low pH at sites that otherwise rebound quickly following stimulation of salivary flow[13,50]. We hypothesize that normal salivary flow gives rise to habitat heterogeneity between sites (i.e., the molar and incisors appear to be different) by generating a pH differential. We further postulate that a clinically significant reduction in salivary flow leads to a reduced difference in oral clearance and intra-plaque pH across the anterior–posterior dimension leading to homogenization of intra-oral compartments and reduced heterogeneity of microbial communities. In keeping with this, we and others[27–30] have found individuals with low salivary flow experience a higher burden of anterior caries and an enrichment of acid-loving, acid-producing organisms even though only one of our participants had active caries at the time of sample collection.

Disease progression and autoimmune processes[51] in SS may explain the varying degree of gradient modification observed between SS patients. Sjögren's syndrome is thought to impact the submandibular/sublingual glands earlier during the natural history of disease with later impairment of the parotid gland[52]. As a consequence, patterns of gradient attenuation or amplification may be related to differences in disease progression; this was not directly assessed as a source of the personalized differences reported in this work. Future studies should focus on identifying the degree to which the date of diagnosis and therefore the duration of hyposalivation influence observed patterns not just on dental surfaces, but on soft tissues as well. Difficulties inherent in working with the SS population include uncertainty surrounding the onset of salivary gland pathology due to a prolonged sub-clinical period of disease. Low salivary flow often antedates symptoms and complaints of xerostomia and SS is consequently diagnosed on average 9 years after the first tooth is lost to the disease[53–55]. To disentangle the relative contribution of autoimmune processes relative to the isolated effect of salivary flow, we are currently assessing the extent to which anti-cholinergic medications induce community gradient modifications similar to those reported here.

Limitations of this study include our focus on whole-salivary flow rates rather than measuring the secretory capacity of individual glands. Glandular measurements, although technically challenging, may help explain the differential responses of the low-flow patients in the degree of gradient attenuation or modulation. Similarly, salivary composition may be as important as the flow rate and may contribute to inter-individual responses, but was not examined in this study. Individuals with SS tend to produce lower salivary concentrations of antimicrobial proteins[53] compared to otherwise healthy individuals, which may partially explain why they harbor more acidogenic and aciduric organisms compared to controls. Moreover, current limitations in Food and Drug Administration (FDA)-approved technologies hinder reliable measurements of dental plaque pH in situ (despite our attempts to do so; this study), so we could not associate spatial patterns in community composition with pH directly. Future work should include efforts to measure intra-plaque pH, glandular flow rates, and salivary composition to quantify the effect of each on the structure of bacterial communities. Finally, our reliance on 16S rRNA gene sequencing limits our ability to assess the extent to which dispersal or—any mechanism other than salivary flow—influences the observed spatial patterns. Additional work should evaluate these spatial patterns using strain-resolved metagenomics, as it is likely that dispersal, among other factors, influences community composition in the human oral cavity.

Here, we identified a gradient in community composition that encompasses the entire spatial extent of the oral cavity. Communities inhabiting very distinct habitats conformed to this gradient, which appeared to be modified in individuals with low saliva production. Our work suggests that characterizing the type and extent of spatial patterns in the human microbiota enables mechanistic studies of the processes that generate, maintain, and disrupt those patterns, and their relationship to health and disease.

## Methods

**Scripts and statistical analyses**. All scripts and data needed to replicate these results are provided as Supplementary Data 1–6, and are also provided at the Stanford Digital Repository (https://purl.stanford.edu/xr749qy9885). All statistical analyses were performed in R-3.4.2, unless otherwise specified.

**Human subjects**. Written, informed consent was obtained from each of 31 unique participants (Supplementary Table 1) prior to sample collection in compliance with human subjects protocols approved by the University of California, San Francisco (UCSF) Human Research Protection Program and Institutional Review Board, and the Stanford University Administrative Panels on Human Subjects in Medical Research. Subjects were recruited into four cohorts: (1) 11 healthy adults were recruited into a "discovery cohort"; (2) 7 additional healthy adults were recruited into a "control cohort" for the validation data set; (3) 3 additional healthy adults were recruited into a "mucosal biogeography cohort"; and (4) 10 individuals who experienced low salivary flow due to the autoimmune disorder, SS, were recruited into a "low-flow cohort" for the validation data set. One individual in each of the discovery and validation cohorts also participated in the mucosal biogeography cohort.

Subjects were asked to refrain from eating, drinking, or performing oral hygiene within 2 h of sample collection or oral screening. A calibrated dentist performed a comprehensive dental exam to evaluate the oral and dental health status of each participant, as per our previously published protocol[56]. With one exception, all subjects were free of periodontal or dental disease, and reported being in good general health. One subject in the mucosal biogeography cohort was found to have root caries on the facial surface of tooth 18 and recurrent caries surrounding occlusal restorations on teeth 15 and 30. No subject had used antibiotics in the 6 months preceding enrollment.

**Measurement of salivary flow rates**. UWS-FRs and SWS-FRs were measured for individuals recruited into the healthy control and low-flow cohorts for the validation data set. UWS-FR and SWS-FR were measured over a period of 5 min using standard protocols[57]. A Welch's two-sample $t$ test was used to determine whether UWS-FR or SWS-FR (ml/min) differed significantly between the control ($N = 7$) and low-flow ($N = 10$) individuals.

**Sample collection protocol**. Universal dental numbering was used to reference teeth. For each of nine adults enrolled in the discovery protocol, samples of the buccal and lingual surfaces of all teeth (excluding third molars) were collected by a dentist at the UCSF School of Dentistry on days 1 and 8 of enrollment (Supplementary Table 1). For the remaining two subjects enrolled in the discovery cohort, samples were collected also on days 15, 22, and 29. For each of the six healthy control subjects, supragingival samples of both the buccal and lingual tooth aspects were collected from all teeth (excluding third molars) by a dentist at the UCSF School of Dentistry on days 1, 8, 15, 22, and 29 of enrollment. Samples for one subject in the control cohort were collected by a clinician only on days 1 and 8.

Individuals in the mucosal biogeography and low-flow cohorts were asked to attend the dental clinic at the UCSF School of Dentistry for sample collection of both the buccal and lingual aspects of all teeth (excluding third molars) on 1 day only. In addition to collecting dental plaque from the mucosal biogeography cohort, we performed a comprehensive survey of the oral mucosa, collecting from each participant 55 samples of the AM; 28 of the BM; and 42 of the KG. For each mucosal surface, samples were collected proximal to the nearest tooth, and samples were mapped to the nearest tooth (e.g., the KG adjacent to the buccal surface of tooth 2, tooth 3, tooth 4, and so on). Samples of the buccal-facing AM were taken from the mucogingival junction to the buccal vestibule at each tooth, whereas "lingual AM" samples were collected at the lingual mucogingival junction; samples of the BM were taken from a line bisecting the cheek to the buccal vestibule overlying each tooth; samples of the KG were taken from the gingival margin of each tooth to the mucogingival junction.

Participants in the discovery cohort and healthy controls in the validation cohort were additionally asked to self-collect samples daily using the same sample collection instruments as the clinician on days 1 through 8 or days 2 through 29 of study enrollment. Discovery subjects were instructed to collect samples from 8 index teeth (teeth #3, 8, 9, 14, 19, 24, 25, 30), whereas validation control subjects were instructed to collect samples from 12 index teeth (teeth #3, 6, 8, 9, 11, 14, 19, 22, 24, 25, 27, 30).

**DNA extraction and barcoded sequencing of the 16S rRNA gene**. Genomic DNA was extracted from all samples using the MoBio PowerSoil DNA Isolation kit in either the tube or the plate-based format (products #12888–100 and 12955–4, MoBio, Carlsbad, CA) according to the manufacturer's instructions. In parallel with true samples, extraction controls were processed using the MoBio protocol including either just reagent ($N = 101$) or reagent plus a sterile sample collection instrument ($N = 300$). Using DNA from 1909 samples collected for the initial discovery data set, PCR primers targeting the V4–V5 region of the 16S rRNA gene were used as previously described[58] and amplicons were sequenced in batches of ~400 on the 454 Ti-Pyrosequencing platform. For validation cohort samples as well as samples collected between days 9 and 29 for 2 discovery subjects, the V4 region of the 16S rRNA gene was PCR-amplified with barcoded primers, as previously described[58], pooled in batches of roughly 800 samples per run, and sequenced along with technical controls across 15 lanes of the Illumina HiSeq 2500 platform (University of Illinois Roy J. Carver Biotechnology Center, Urbana, IL). A full description of the demultiplexing and quality-filtering steps used to exclude sequences and construct the taxa by sample data matrices is provided (Supplementary Methods; Supplementary Data 1).

**Examining communities on the molars and incisors**. Since the analysis of the discovery data suggested that communities on molars and incisors differed, we sought to determine whether the effect was similar with samples sequenced more deeply. A data subset consisting of molar and incisor samples ($N = 3393$) collected from eight healthy controls from the validation cohort was analyzed. Taxa ($N = 90$) counts were Hellinger-transformed before PCoA was performed on Bray-Curtis dissimilarity. The robustness of the findings to various transformations and distance metrics was also evaluated (Supplementary Methods; Supplementary Note 1). To assess the relative contribution of the factors, subject, tooth class, and tooth aspect on community structure, an analysis of dissimilarity (Adonis) was performed on Bray Curtis dissimilarity. To account for the non-independence of temporal replicates, we summed across time points within each individual before performing Adonis using the vegan package[59] of R. Permutations of the species by sample data matrix, during Adonis, were restricted within individuals.

**Modeling geographic coordinates of sample sites**. The pixel coordinates $(x, y)$ for each sample site were modeled using WebPlotDigitizer[60] trained on an image of the oral cavity.

**Evaluating conformation of communities to a gradient**. To evaluate whether microbial communities inhabiting supragingival surfaces are structured along a gradient, a data set consisting of 117 taxa across 1701 samples collected from all teeth (excluding third molars) from each of 7 healthy adults was evaluated using a trend surface analysis[61]. A PCA using the dudi.pca function of the ade4 package[62] was used to obtain a duality diagram for subsequent input into a PCA with respect to instrumental variables (PCA-IV) in which a third-order polynomial function of the modeled geographic coordinates was used as the constraint. To examine whether mucosal communities also conformed to a gradient, a trend surface analysis was also performed on the mucosal biogeography data set (542 samples; 101 taxa) consisting of buccal and lingual samples of supragingival plaque, and as applicable, of the BM, AM, and KG collected from three additional individuals. The robustness of these findings was assessed by comparing the results to those of a PCNM[63] and by analysis using MEM and by EMS (Supplementary Notes 4–5; Supplementary Data 4).

**Impact of clinical variables on community composition**. Constrained correspondence analysis was used to evaluate the extent to which UWS-FR, SWS-FR, and the MFS index for smooth surfaces (Supplementary Methods) explain variation in community composition across supragingival surfaces. The validation data set consisting of samples from the low-flow and healthy control cohorts (147 taxa; 825 samples) was used in the analysis. Permutational analysis of variance was used to assess the significance of UWS-FR, SWS-FR, and MFS as predictors of community composition with all permutations stratified within subjects. To evaluate the taxa that explain the segregation of samples into low-flow and healthy control groups, we projected the taxa onto the first and second coordinates.

**Data availability**. The data supporting the results of this study are available in the NIH Short Read Archive, SRA accession number SRP126946: (http://www.ncbi.nlm.nih.gov/sra). The code and data that were used to generate these findings can also be found at: https://purl.stanford.edu/xr749qy9885. All other data supporting the findings of this study are available within the article and its Supplementary Information files, or are available from the authors upon request.

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

## Acknowledgements

We thank the volunteers who participated in this study, as well as colleagues who provided technical assistance or instruction, including Les Dethlefsen, Elizabeth Costello, Denise Monack, Manuel Amieva, Elisabeth Bik, Shanne Sastiel, Tina Jung, Jonathan Weng, Lisa Jack, Swetha Kanukula, and Nancy Crabbe. We are especially grateful to the SS Foundation. This work was supported by the National Institutes of Health (DP1OD000964 and R01DE023113 to D.A.R.), and by the Thomas C. and Joan M. Merigan Endowment at Stanford University (D.A.R.). D.M.P. was supported by a Stanford Graduate Fellowship through the Office of the Provost for Graduate Education and by the Cellular and Molecular Biology (5T32GM007276) and Molecular Basis of Host Parasite Interactions (5T32AI007328) training grants to Stanford University.

## Author contributions

All authors agree to submission of this manuscript and are responsible for the accuracy, integrity and ethics of the whole or of the methods or data they contributed. D.M.P, P.M.L., G.C.A., S.A.L., S.P.H., and D.A.R. contributed to experimental design and developed the IRB protocols. D.M.P. and N.M.D. prepared samples for sequencing. D.M.P., J.A.F., and S.P.H. analyzed and interpreted the data with input from D.A.R. D.M.P. wrote and revised the manuscript, which was edited for content and clarity by D.A.R. and approved in its final form by all co-authors.

## Additional information

**Competing interests:** The authors declare no competing financial interests.

