## [Peer Review File · Nature Communications]

Reviewers' comments:

Reviewer #1 (Remarks to the Author):

The investigators present extensive data on the spatial patterning of bacterial diversity in the oral cavity. The manuscript is very well written and presents informative figures both in the body of the manuscript and in supplementary materials. In a previous review of an earlier version of this manuscript, this reviewer was concerned with the conclusion that microbial gradients are shaped predominantly by salivary flow. The possibility that known temperature gradients could provide an alternative explanation is now acknowledged (line 317), but then dismissed in a one sentence discussion. The findings of the most pertinent reference in the older literature are not cited or discussed.

Haffajee AD, Socransky SS, Smith C, Dibart S, Goodson JM. 1992. Subgingival temperature (III). Relation to microbial counts. *J Clin Periodontol.* 19(6):417-22. PMID: 1321846.

While this reviewer believes that the inclusion of temperature data would have strengthened study and possibly led to more nuanced conclusions on the role of salivary flow in determining microbial gradients, the data present is novel and compelling in defining the spatial diversity of the microbiota on various oral surfaces. The approaches taken and analytical methods are a model for the study of not only the oral cavity, but also for the study microbial gradients in other host or environmental sites.

Statistical methods appear appropriate and are well documented in the supplementary materials where scripts for generation of figures are fully presented.

This reviewer recommends acceptance of the manuscript in its current form.

Reviewer #2 (Remarks to the Author):

This manuscript describes an interesting study in which the authors make use of spatial ecology techniques to assess the spatial and environmental factors controlling the composition of microbial communities in the human oral cavity. The study also makes an interesting comparison between the spatial distribution of microbial communities in the oral cavities of healthy subjects compared with what was found in the oral cavities of SS patients.

I am an ecologist with experience using spatial and multivariate stats so I will focus my comments on the spatial ecology and multivariate statistics aspects of the paper, and defer to other reviewers about issues related to dentistry or microbiology lab techniques and bioinformatics.

Generally, this is a well written manuscript and represents a novel application of techniques typically used in ecology to understand dynamics influencing the microbiome in the human oral cavity. This sort of cross-discipline thinking is really healthy for science in general. I think the authors should update some of the multivariate/spatial statistics techniques they used in this study, and if the patterns qualitatively remain the same, then I would recommend the paper be accepted.

The authors test hypotheses about the relative importance of local "environmental factors" versus spatial influences (typically related to dispersal of microbes across microhabitats within the oral cavity). The authors hypothesize that the production of saliva provides a vector for dispersal, thus, increased saliva production is associated with increased microbial dispersal within the oral cavity, potentially creating more homogeneity in microbial composition across sites.

In the ecological literature, there are "paradigms" that could be used to describe the different dynamics that can arise to structure the microbial assemblages under these different conditions (see Leibold et al. 2004 in *EcoLetters*). For example, the Baas-Becking type dynamics (Everything is everywhere, but, the environment selects) is called "species sorting". A situation where

increased saliva production promotes the dispersal of microbes from source communities to sink communities along a gradient could be referred to as "mass effects". Leibold et al, and later Logue et al. (2011 in TREE) describe how these "metacommunity" dynamics can emerge depending on the level of dispersal and the degree of habitat heterogeneity. This literature provides the foundation for how many ecologists – at present – interpret the sort of analyses the authors are using.

Leibold, M. A., M. Holyoak, N. Mouquet, P. Amarasekare, J. M. Chase, M. F. Hoopes, R. D. Holt, J. B. Shurin, R. Law, D. Tilman, M. Loreau, and A. Gonzalez. 2004. The metacommunity concept: a framework for multi-scale community ecology. *Ecology Letters* 7:601–613.

Logue, J. B., N. Mouquet, H. Peter, H. Hillebrand, M. W. Group, and others. 2011. Empirical approaches to metacommunities: a review and comparison with theory. *Trends in ecology & evolution* 26:482–491.

Regarding the analyses that were used. There are a few issues that should be addressed – but I don't think they're fatal flaws for the paper.

(1) The authors rely too much on scores for axis 1 from the PCoA. There multivariate methods that are analogous to most if not all of the analyses that the authors used in this manuscript. Reducing the response variable down to a univariate metric, such as a PCoA axis, is not necessary, and can be problematic when trying to interpret community turnover along a gradient. Specifically, gradients in community composition usually do not manifest as a linear trajectory in an ordination plot. Rather, they usually follow a "horseshoe" trajectory in 2-D ordination space. The consequence is that only looking at scores along one axis can be misleading if both ends of the compositional gradient occur on the same end of the axis of interest – think a parabola of $y^2 = \text{PCo axis 1 scores}$, where y is your "community composition". You could have similar axis scores for communities at different ends of the "community composition gradient". I suggest looking at McCune and Grace 2001 (the PCOrd book) and or see Legendre and Gallagher (2001 in *Oecologia*, Figure 4). The patterns that the authors identified may very well be real, but the possibility exists that they are picking up a statistical artefact using only Axis 1 scores.

Legendre, P., and E. D. Gallagher. 2001. Ecologically meaningful transformations for ordination of species data. *Oecologia* 129:271–280.

McCune, B., and J. B. Grace. 2002. *Analysis of Ecological Communities*. MjM Software Design, Gleneden Beach, OR.

(2) I think the Elements of Metacommunity Structure analysis is very well suited to inform some of the ecological questions the authors are asking in this manuscript. This analysis is very well suited for characterizing how community composition shifts along a gradient (e.g., species turnover vs. gain of species, etc.). This would aid in the ecological interpretation of turnover from the front to the back of the oral cavity, and might provide some insight into the mechanism (species replacement, vs. mass effects with the flow of saliva). A good application of this technique can be found in Willig and Presley (2015 in the *Journal of Tropical Ecology*, see Figure 3). The metacom package for R has the functions necessary for these sorts of analyses.

Willig, M. R., and S. J. Presley. 2015. Biodiversity and metacommunity structure of animals along altitudinal gradients in tropical montane forests. *Journal of Tropical Ecology FirstView*:1–16.

(3) The authors should update their method for modelling spatial heterogeneity of the microbial community. The more commonly used method in the current literature is called Moran Eigenvector Map (MEM) analysis – I believe this can be done with the `vegan::pcnm` function – the authors cite

the vegan package for R, so I assume it would not be too difficult for them to implement this analysis. The `pcnm` function was originally written to conduct a special case of MEM analysis called principal coordinate of neighbor matrices, but my understanding is that the function (still called `pcnm`) has been updated for the more current version of MEM analysis. This analysis creates "spatial filters" which represent different scales of heterogeneity that can then be used as predictors in a CCA or RDA – similar to what the authors did, but this is considered a more comprehensive approach to modeling spatial patterns for this sort of analysis. See Dray et al. 2012.

Dray, S., R. Péliissier, P. Couteron, M.-J. Fortin, P. Legendre, P. R. Peres-Neto, E. Bellier, R. Bivand, F. G. Blanchet, M. De Cáceres, A.-B. Dufour, E. Heegaard, T. Jombart, F. Munoz, J. Oksanen, J. Thioulouse, and H. H. Wagner. 2012. Community ecology in the age of multivariate multiscale spatial analysis. *Ecological Monographs* 82:257–275.

Fig 1. Univariate tests on scores from one axis in an ordination are not the standard method for evaluating differences in community composition. Especially since axis 1 only explains ~25% of the total variation in community composition. The authors are clearly familiar with the vegan package for R (citation 62). This package has multivariate methods that are better suited for multivariate test for differences in community composition. The authors do make use of the `adonis()` function, which I think is a better approach for showing differences among groups. I think the authors should remove panel D from the figure. They should instead just report results from the Adonis analysis as thier test for differences in community composition between Incisor and molar communities in Buccal and Lingual aspects. In this sort of permutational test, I would constrain permutations to within-subjects to better account for random effects among subjects.

The subsequent figures may change if the authors follow some of my recommendations above.

Point-by-point response to reviewer comments for NCOMMS-17-18865

Reviewer comments are displayed in boldface font. Author responses are indicated beneath the reviewer comment boxes. All page and line numbers refer to the revised text.

Reviewer #1 (Remarks to the Author):

#1: Reviewer 1 Comment 1: The investigators present extensive data on the spatial patterning of bacterial diversity in the oral cavity. The manuscript is very well written and presents informative figures both in the body of the manuscript and in supplementary materials. In a previous review of an earlier version of this manuscript, this reviewer was concerned with the conclusion that microbial gradients are shaped predominantly by salivary flow. The possibility that known temperature gradients could provide an alternative explanation is now acknowledged (line 317), but then dismissed in a one sentence discussion. The findings of the most pertinent reference in the older literature are not cited or discussed.

Haffajee AD, Socransky SS, Smith C, Dibart S, Goodson JM. 1992. Subgingival temperature (III). Relation to microbial counts. J Clin Periodontol. 19(6):417-22. PMID: 1321846.

#1: We thank reviewer 1 for these comments and appreciate having this important work by Haffajee et al. brought to our attention. In the first draft, we had cited a primary paper from 1978 that antedates this work by Haffajee et al. (1992). After further review of the Haffajee et al. paper (1992) referenced by reviewer 1, we have added it to our citations. While Brill et al. was, to our knowledge, the first to empirically identify temperature as a variable that may structure communities in the subgingival environment, Haffajee et al. examined both temperature and its impact of the communities of the subgingival crevice. Lines pertaining to temperature were otherwise unchanged (Lines 416-421). We also note that our work here has focused on the supragingival communities, and not those of the subgingival crevice where temperature was shown to differ and may play a different role than in the supragingival context.

#2. Reviewer 1 Comment 2: While this reviewer believes that the inclusion of temperature data would have strengthened study and possibly led to more nuanced conclusions on the role of salivary flow in determining microbial gradients, the data present is novel and compelling in defining the spatial diversity of the microbiota on various oral surfaces. The approaches taken and analytical methods are a model for the study of not only the oral cavity, but also for the study microbial gradients in other host or environmental sites.

#2: We agree that an investigation of temperature would provide additional insight into the ecology of the human oral cavity, but collecting temperature measurements would increase dental chair time beyond the current duration of 1-1.5 hours (per sample collection

appointment). Early on in this set of experiments, we collected a variety of tangential samples, including subgingival samples, resulting in per-appointment chair times in excess of 2-2.5 hours. In our experience, limiting dental chair time to the current duration of 1-1.5 hours is crucial for the successful enrollment and retention of human subjects. To probe and record temperature and to collect subgingival samples in addition to the current protocol would prevent adequate progress towards our primary aims – investigating the role of salivary flow in structuring oral microbial communities. For this reason, we believe that an investigation of temperature would more appropriately fit into a focused study that specifically aims to elucidate the effects of temperature on microbial communities across the anterior to posterior dimension. It is our hope that by providing all the code needed to reproduce our findings to the scientific community other investigators may be enabled to undertake an investigation geared towards answering questions such as this when analyzing the spatial variation of communities.

#3. Reviewer 1 Comment 3: Statistical methods appear appropriate and are well documented in the supplementary materials where scripts for generation of figures are fully presented.

This reviewer recommends acceptance of the manuscript in its current form.

#3: We thank the reviewer for this assessment.

Reviewer #2 (Remarks to the Author):

#4. Reviewer 2 Comment 1: This manuscript describes an interesting study in which the authors make use of spatial ecology techniques to assess the spatial and environmental factors controlling the composition of microbial communities in the human oral cavity. The study also makes an interesting comparison between the spatial distribution of microbial communities in the oral cavities of healthy subjects compared with what was found in the oral cavities of SS patients.

I am an ecologist with experience using spatial and multivariate stats so I will focus my comments on the spatial ecology and multivariate statistics aspects of the paper, and defer to other reviewers about issues related to dentistry or microbiology lab techniques and bioinformatics.

Generally, this is a well written manuscript and represents a novel application of techniques typically used in ecology to understand dynamics influencing the microbiome in the human oral cavity. This sort of cross-discipline thinking is really healthy for science in general. I think the authors should update some of the multivariate/spatial statistics techniques they used in this study, and if the patterns qualitatively remain the same, then I would recommend the paper be accepted.

#4: We appreciate these comments about our study and suggestions for additional analyses. We have acted upon these suggestions and undertaken additional analyses. We believe the reviewer will find, as outlined below, that the patterns remain the same.

#5. Reviewer 2 Comment 2: The authors test hypotheses about the relative importance of local “environmental factors” versus spatial influences (typically related to dispersal of microbes across microhabitats within the oral cavity). The authors hypothesize that the production of saliva provides a vector for dispersal, thus, increased saliva production is associated with increased microbial dispersal within the oral cavity, potentially creating more homogeneity in microbial composition across sites.

#5: Rather than proposing a role for dispersal in structuring the gradient, we hypothesize that normal salivary flow gives rise to habitat heterogeneity between sites (i.e., the molar and incisors appear to be different) by generating a pH differential. This hypothesis is articulated in the main manuscript (Lines 423-441). The absence of saliva appears to attenuate the pH differential¹, resulting in the homogenization of site-to-site heterogeneity (i.e., the relative difference between the molars and incisors is reduced). We believe that pH, rather than dispersal, is driving this effect since a reduction in salivary flow results in the homogenization of intra-plaque pH across the anterior-posterior dimension by reducing oral clearance². In keeping with this, we observed an enrichment in acid-loving/acid-producing organisms in the mouths of individuals with low salivary flow.

Given the paucity of FDA approved pH probes for measuring plaque pH *in situ* in humans, and the issues we had with the WPI Beetrode dental electrodes, which were pulled from the

market, for a time, as a result of the probes falling off in patients' mouths, it was not possible for us to test this hypothesis explicitly. Nonetheless, it is our belief that the elucidation of a spatial pattern in a host-associated community and the elucidation of the spatial extent of that pattern is sufficiently novel to warrant publication of this work.

As an aside, one difficulty inherent in the use of the V4 hypervariable region of the 16S rRNA for study of the oral microbiota is its inability to resolve the taxonomy of most organisms to the strain level. Without strain-level information, it is not possible to do a study of dispersal. We now acknowledge this limitation in the discussion (Lines 471-475). Without strain-resolved sequencing our ability to make inferences about the extent that dispersal plays in shaping community composition is limited. For this reason, we are pursuing the strain-resolved metagenomics sequencing of a subset of samples collected over time in order to determine whether the strains at neighboring sites are more or less similar to each other than to strains at distal sites. Such an undertaking warrants an independent investigation – the purpose of the current work was to determine whether or not sites in the oral cavity conformed to a random spatial distribution or on the contrary whether sites exhibited non-random patterns of variation across space – we believe the elucidation of this pattern is the first step towards testing mechanisms that give rise to patterning.

#6. Reviewer 2 Comment 3: In the ecological literature, there are “paradigms” that could be used to describe the different dynamics that can arise to structure the microbial assemblages under these different conditions (see Leibold et al. 2004 in *EcoLetters*). For example, the Baas-Becking type dynamics (Everything is everywhere, but, the environment selects) is called “species sorting”. A situation where increased saliva production promotes the dispersal of microbes from source communities to sink communities along a gradient could be referred to as “mass effects”. Leibold et al, and later Logue et al. (2011 in *TREE*) describe how these “metacommunity” dynamics can emerge depending on the level of dispersal and the degree of habitat heterogeneity. This literature provides the foundation for how many ecologists – at present – interpret the sort of analyses the authors are using.

- Leibold, M. A., M. Holyoak, N. Mouquet, P. Amarasekare, J. M. Chase, M. F. Hoopes, R. D. Holt, J. B. Shurin, R. Law, D. Tilman, M. Loreau, and A. Gonzalez. 2004. The metacommunity concept: a framework for multi-scale community ecology. *Ecology Letters* 7:601–613.
- Logue, J. B., N. Mouquet, H. Peter, H. Hillebrand, M. W. Group, and others. 2011. Empirical approaches to metacommunities: a review and comparison with theory. *Trends in ecology & evolution* 26:482–491.

#6: We have assessed community coherence, turnover and clumping using Metacommunity Theory, as suggested by the reviewer. Our data (new Figure S14) revealed communities in both healthy controls and in patients with SS exhibit coherence, less turnover than expected based on chance, and more clumping than expected by chance. Taken together, these results suggest that supragingival plaque biofilms may be described as conforming to a nested

structure with clumped species loss across the anterior-posterior dimension, as discussed (Lines 379-383).

In addition, we now acknowledge as a limitation of the current work that it is beyond the resolution of 16S rRNA gene sequencing, using the V4 region, to assess the extent to which dispersal plays a role in generating the observed spatial pattern. We are currently undertaking work to examine the role that dispersal plays in shaping spatial patterns in the oral cavity using strain-resolved metagenomics.

The purpose of this manuscript was to determine whether it was possible to identify a spatial pattern present in the human microbiome. Upon finding one, we tested a specific hypothesis about the role that salivary flow plays in maintaining the spatial pattern. It is beyond the scope of the current work to test every possible mechanism which may give rise to the observed gradient. The framework provided here to identify spatial patterning in the context of the human microbiome is itself significant.

#7. Reviewer 2 Comment 4: Regarding the analyses that were used. There are a few issues that should be addressed – but I don't think they're fatal flaws for the paper.

(1) The authors rely too much on scores for axis 1 from the PCoA. There multivariate methods that are analogous to most if not all of the analyses that the authors used in this manuscript. Reducing the response variable down to a univariate metric, such as a PCoA axis, is not necessary, and can be problematic when trying to interpret community turnover along a gradient. Specifically, gradients in community composition usually do not manifest as a linear trajectory in an ordination plot. Rather, they usually follow a “horseshoe” trajectory in 2-D ordination space. The consequence is that only looking at scores along one axis can be misleading if both ends of the compositional gradient occur on the same end of the axis of interest – think a parabola of $y^2 = \text{PCO axis 1 scores}$, where y is your “community composition”. You could have similar axis scores for communities at different ends of the “community composition gradient”. I suggest looking at McCune and Grace 2001

(the PCOrd book) and or see Legendre and Gallagher (2001 in *Oecologia*, Figure 4). The patterns that the authors identified may very well be real, but the possibility exists that they are picking up a statistical artefact using only Axis 1 scores.

Legendre, P., and E. D. Gallagher. 2001. Ecologically meaningful transformations for ordination of species data. *Oecologia* 129:271–280.

McCune, B., and J. B. Grace. 2002. *Analysis of Ecological Communities*. MjM Software Design, Gleneden Beach, OR.

#7: Thank you for highlighting these issues. In response, we note two key points.

1. We did not include an interpretation of other axes in order to streamline the story, keeping

the focus of the manuscript on the major component of variation and the identification of the anterior-posterior gradient. Nonetheless, we appreciate this alternative perspective. As a consequence, we have supplemented the figures in the main manuscript with the interpretation of other ordination axes:

- a) Using a PCoA on Bray Curtis computed on Hellinger-transformed data (new Figure 1), we concluded that molar and incisor communities differ along axis 1. Now, we present an analysis of all pairwise combinations of the first, second, third and fourth ordination axes (new Fig. S1; Supplementary Results and Discussion Lines 309-317). Figure S1 reveals that molar and incisor samples differentiate not along the second, third or fourth axes, but only along the first – particularly if samples originated at the lingual tooth aspect. Further, none of the sample distributions appear to conform to a parabolic or horseshoe shape, suggesting that interpretation of axis 1 scores makes sense.
 - b) Using a trend surface analysis of Hellinger-transformed data (new Figure 2) we concluded that the molars and incisors represent opposing poles of a gradient in community composition with the remaining tooth classes arrayed in an ordered, structured fashion in between. We have now added a forward selection analysis of the polynomial terms and analyzed the 4 significant RDA axes as well as 6 significant polynomial terms (new Figure S7; Supplementary Results & Discussion Lines 354-387). By projecting the sample scores onto a map of the mouth, we are able to identify patterns in community composition that were not discussed in the original draft: the segregation of communities by tooth aspect and by jaw was discovered as a significant spatial variable in addition to variance across the anterior-posterior dimension. The first and second axes were also plotted against each other and the data do not conform to a parabolic shape (modified Supplementary Data File 2).
 - c) Using a trend surface analysis we concluded that mucosal communities also conform to an anterior to posterior gradient (new Figure 3). To supplement the interpretation of the first axis, we have now added a forward selection analysis of the polynomial terms and used a permutational ANOVA to test the RDA axes for significance. All significant axes are now interpreted in the supplemental discussion (new Figure S11; Supplementary Discussion Lines 477-498).
 - d) We now interpret all significant axes for the PCNM and MEM analyses that were added (new Figures S8-S9; Supplementary Discussion Lines 389-475; Supplementary Data File 4).
2. In addition, we have taken the reviewer’s suggestion, as implied by the inclusion of the Legendre & Gallagher (2001) citation, to evaluate the robustness of the observed findings to various “ecologically meaningful” transformations, comparing the Hellinger and Chord transformations to the variance stabilizing (VST) and relative abundance transformations typically used by investigators in the field of microbiome research who work with sequence count data. This analysis demonstrates that the separation of the molar and incisor communities is robust to a variety of transformation (new Fig. S2 Supplementary Discussion Lines 309-317). Even so, rather than universally adopting the VST transformation we have performed the majority of analyses on Hellinger transformed data. We note that the results from the Hellinger transformed data are qualitatively the same as the findings generated from VST-transformed data.

#8. Reviewer 2, Comment 5: (2) I think the Elements of Metacommunity Structure analysis is very well suited to inform some of the ecological questions the authors are asking in this manuscript. This analysis is very well suited for characterizing how community composition shifts along a gradient (e.g., species turnover vs. gain of species, etc.). This would aid in the ecological interpretation of turnover from the front to the back of the oral cavity, and might provide some insight into the mechanism (species replacement, vs. mass effects with the flow of saliva). A good application of this technique can be found in Willig and Presley (2015 in the Journal of Tropical Ecology, see Figure 3). The metacom package for R has the functions necessary for these sorts of analyses.

Willig, M. R., and S. J. Presley. 2015. Biodiversity and metacommunity structure of animals along altitudinal gradients in tropical montane forests. Journal of Tropical Ecology FirstView:1–16.

#8: We appreciate the reviewer's suggestion to use the metacom package. We have used the R implementation (new Supplementary Data File 5)³ to examine the coherence (new Fig. S14a), turnover (new Figs. S14b/S14e) and boundary clumping (new Figs. S14c/S14f) of communities across the anterior to posterior gradient. Metacommunity analysis identified the spatial pattern as one of nestedness with clumped species loss across the anterior-posterior dimension as now discussed in the results of the main text (Lines 379-383) and in the Supplementary Discussion (Supplementary Discussion, Lines 530-602).

Intriguingly, though we did not quantify these differences, we have observed that communities in the oral cavities of patients with low salivary flow due to SS tended to show reduced turnover and heightened clumping as compared to communities in the mouths of healthy controls.

#9. Reviewer 2, Comment 6: (3) The authors should update their method for modelling spatial heterogeneity of the microbial community. The more commonly used method in the current literature is called Moran Eigenvector Map (MEM) analysis – I believe this can be done with the `vegan::pcnm` function – the authors cite the `vegan` package for R, so I assume it would not be too difficult for them to implement this analysis. The `pcnm` function was originally written to conduct a special case of MEM analysis called principal coordinate of neighbor matrices, but my understanding is that the function (still called `pcnm`) has been updated for the more current version of MEM analysis. This analysis creates “spatial filters” which represent different scales of heterogeneity that can then be used as predictors in a CCA or RDA – similar to what the authors did, but this is considered a more comprehensive approach to modeling spatial patterns for this sort of analysis. See Dray et al. 2012.

Dray, S., R. Pélissier, P. Coutron, M.-J. Fortin, P. Legendre, P. R. Peres-Neto, E. Bellier, R. Bivand, F. G. Blanchet, M. De Cáceres, A.-B. Dufour, E. Heegaard, T.

Jombart, F. Munoz, J. Oksanen, J. Thioulouse, and H. H. Wagner. 2012. Community ecology in the age of multivariate multiscale spatial analysis. *Ecological Monographs* 82:257–275.

#9: To increase the robustness of the trend surface analyses, we performed forward selection analysis of the polynomial terms (new Figure S7; new Figure S10; Supplementary Results & Discussion Lines 354-387; 477-498). In addition, a common complaint about the trend surface analysis is that it is unable to detect fine scale spatial patterns, examining instead only broad scale spatial variables⁴. As suggested by Reviewer 2, a class of spatial analyses known as Moran's Eigenvector Maps (MEM) circumvents this problem, capably identifying both fine and broad scale spatial patterns.

In the first MEM analysis, we performed a Principal Components Analysis of Neighbor Matrices (PCNM) where the neighborhood was defined as the minimal distance giving rise to a connected network (new Figure S8; Supplementary Results & Discussion Lines 389-426). This analysis again revealed the anterior to posterior gradient as a significant spatial structure. In addition, it gave rise to a variety of significant fine scale variables that were difficult to interpret, raising the question as to how a neighborhood should be defined given that the distance threshold used to define a neighbor was in many ways arbitrarily selected.

To avoid the use of such an arbitrary threshold we also performed a second MEM analysis, constructing a set of 20 different neighbor matrices each differing from the others by an equal distance between 0.2 and 2 (new Figure S9; Supplementary Results & Discussion Lines 428-475). We compared the performance of each of the 20 models for their ability to explain community composition, evaluating the models using the Akaike information criterion (AIC). This MEM analysis also revealed the anterior to posterior gradient to be a significant spatial structure, in addition to identifying a variety of other significant fine and broad scale spatial patterns.

Importantly, in all cases – the trend surface analysis, the PCNM with an arbitrary threshold, and the MEM analysis involving model selection – the difference between the anterior and posterior mouth corresponded to a significant RDA axis and a significant explanatory variable. We have chosen to retain the trend surface analysis in the main text because it is easier to interpret.

#10. Reviewer 2 Comment 7: Fig 1. Univariate tests on scores from one axis in an ordination are not the standard method for evaluating differences in community composition. Especially since axis 1 only explains ~25% of the total variation in community composition. The authors are clearly familiar with the vegan package for R (citation 62). This package has multivariate methods that are better suited for multivariate test for differences in community composition. The authors do make use of the adonis() function, which I think is a better approach for showing differences among groups. I think the authors should remove panel D from the figure. They should instead just report results from the Adonis analysis as thier test for differences in community

composition between Incisor and molar communities in Buccal and Lingual aspects. In this sort of permutational test, I would constrain permutations to within-subjects to better account for random effects among subjects.

#10: We appreciate this suggestion. We have deleted the Wilcoxon Rank Sum test from the analysis in Figure 1. However, we have chosen to retain panel D of the figure since examination of all pairwise plots between the first, second, third and fourth coordinates suggests that communities segregate across axis 1, and they do not conform to a parabola shape. The display of data in this way helps viewers to see the relationship between tooth aspect and tooth class in a way that the results of a permutational anova do not.

We note that in the original manuscript and in the current version as well, the permutations performed with the Adonis function were stratified within subjects (Supplementary Data File 2, section 1f).

#11. Reviewer 2 Comment 8: The subsequent figures may change if the authors follow some of my recommendations above.

We have a number of figures to the supplementary materials, choosing to retain the original structure of the manuscript since the new analyses support and supplement the original findings. Moreover, we believe it is important for readers who are not familiar with spatial statistics to be presented with a single pattern in the main text, highlighting the retention of that pattern in oral mucosal sites, and the attenuation of that pattern under conditions of low salivary flow. We feel that adding these additional figures to the main manuscript would be confusing for many readers.

References

1. Englander HR. The effects of saliva on the pH and lactate concentration in dental plaques. I. Caries-rampant individuals. *Journal of dental research* **38**, 848-853 (1959).
2. Dawes C. A mathematical model of salivary clearance of sugar from the oral cavity. *Caries research* **17**, 321-334 (1983).
3. Dallas T. metacom: an R package for the analysis of meta- community structure. *Ecography* **37**, (2014).
4. Daniel Borcard FG, Pierre Legendre. *Numerical Ecology with R*. Springer (2011).

REVIEWERS' COMMENTS:

Reviewer #2 (Remarks to the Author):

Note: I don't think the line numbers in the manuscript version I'm looking at match what the authors are using in their responses.

I think this manuscript should be accepted with some very minor edits suggested below.

Response #5 – Reviewing my comment and the authors' response, I must have misunderstood the hypothesis when I first reviewed this manuscript. The authors' explanation of the effect of the pH over dispersal makes sense, and fits their ecological explanation as described in the manuscript. However, the specific manuscript lines referenced (423-441) in this response do not clearly articulate this hypothesis – perhaps the authors meant lines 323-341?

Response #6 – see my comments for line 296 below.

Response #7 – The authors were very thorough in addressing these issues, and their analyses seem quite robust.

Response #8 – The authors appropriately used this method, but see comment for line 296 below.

Response #9 – The authors show very robust analyses supporting their statements about the difference between the anterior and posterior mouth

Lines 155-161 – The authors should report the p-values associated with the Adonis models used to calculate the partial R² statistics that are reported.

Lines 214-223 – This result seems quite robust

Line 296 – I think EMS stands for "Elements of Metacommunity Structure" – authors should double check.

Line 296 – I had suggested the authors conduct this analysis. If it is included in the final version of the manuscript, they should include an interpretation of how it supports their conclusions in the discussion, otherwise, it seems out of place. Specifically, this analysis seems to suggest that turnover in bacterial community composition along the anterior-posterior dimension is a result of species loss, rather than species replacement. Does this inform the discussion of the hypothesized mechanism? In particular, this could be used as evidence to support the statement made on line 320.

Point-by-point response to reviewer comments for NCOMMS-17-18865

Reviewer comments are displayed in boldface font. Author responses are indicated beneath the reviewer comment boxes. All page and line numbers refer to the revised text.

REVIEWERS' COMMENTS:

Reviewer #2 (Remarks to the Author):

Note: I don't think the line numbers in the manuscript version I'm looking at match what the authors are using in their responses.

We apologize for this inconvenience.

I think this manuscript should be accepted with some very minor edits suggested below.

We thank the reviewer for this assessment.

Response #5 – Reviewing my comment and the authors' response, I must have misunderstood the hypothesis when I first reviewed this manuscript. The authors' explanation of the effect of the pH over dispersal makes sense, and fits their ecological explanation as described in the manuscript. However, the specific manuscript lines referenced (423-441) in this response do not clearly articulate this hypothesis – perhaps the authors meant lines 323-341?

We apologize for this inconvenience. The correct line numbers for the revised main manuscript are in the fourth paragraph of the discussion (current Lines 530-542). The text has also been pasted below to avoid further line number collisions.

In healthy humans, salivary film velocity and oral clearance rates vary between the front and the back of the mouth^{1,2}. As a consequence, in healthy humans the duration of plaque exposure to dietary and microbial-derived acids, and hence pH, varies across the anterior to posterior dimension– and, a reduction in salivary flow results in prolonged periods of low pH at sites that otherwise rebound quickly following stimulation of salivary flow^{3,4}. We hypothesize that normal salivary flow gives rise to habitat heterogeneity between sites (i.e., the molar and incisors appear to be different) by generating a pH differential. We further postulate that a clinically significant reduction in salivary flow leads to a reduced difference in oral clearance and intra-plaque pH across the anterior-posterior dimension leading to homogenization of intra-oral compartments and reduced heterogeneity of microbial communities. In keeping with this, we and others^{5,6,7,8} have found individuals with low salivary flow experience a higher burden of anterior caries and an enrichment of acid-loving, acid-producing organisms even though none of our participants had active caries at the time of sample collection.

Response #6 – see my comments for line 296 below.

See the response to line 296 below – we appreciate the reviewer for introducing us to the analysis of elements of metacommunity structure.

Response #7 – The authors were very thorough in addressing these issues, and their analyses seem quite robust.

We thank the reviewer for the suggestion to include other ordination axes and data transformations in the manuscript.

Response #8 – The authors appropriately used this method, but see comment for line 296 below.

See the response to line 296 below.

Response #9 – The authors show very robust analyses supporting their statements about the difference between the anterior and posterior mouth

We thank the reviewer for the suggestion to include the Principal Components Analysis of Neighbor Matrices in the analysis.

Lines 155-161 – The authors should report the p-values associated with the Adonis models used to calculate the partial R² statistics that are reported.

We now report the p-values associated with the Adonis model in Supplementary Note 1. Since later analyses reveal the molars and incisors to represent parts of a spatial series we caution against interpreting the p-values. The non-independence of data in a spatial series artificially inflates the probability of a false positive, which makes the interpretation of p-values difficult. Even so, the effect sizes that are assessed by use of Adonis allow for the estimation of the relative contribution of each factor to the structure of the data.

Lines 214-223 – This result seems quite robust

We thank the reviewer for this assessment.

Line 296 – I think EMS stands for “Elements of Metacommunity Structure” – authors should double check.

We have double-checked and corrected the text from “Empirical Metacommunity Structure” to “Elements of Metacommunity Structure”, thanks for catching this.

Line 296 – I had suggested the authors conduct this analysis. If it is included in the final version of the manuscript, they should include an interpretation of how it supports their conclusions in the discussion, otherwise, it seems out of place. Specifically, this analysis seems to suggest that turnover in bacterial community composition along the anterior-posterior dimension is a result of species loss, rather than species replacement. Does this

inform the discussion of the hypothesized mechanism? In particular, this could be used as evidence to support the statement made on line 320.

The referenced text has been moved to the supplement. We believe that EMS analysis does add value to the manuscript, but think that any conclusions drawn from the analysis should be viewed as exploratory until additional approaches can be undertaken to test the robustness and stability of the findings.

We have added an interpretation of the results of the current EMS analysis as described in the revised Supplementary Note 8 (Lines 363-382).

References

1. Dawes C, Watanabe S, Biglow-Lecomte P, Dibdin GH. Estimation of the velocity of the salivary film at some different locations in the mouth. *Journal of dental research* **68**, 1479-1482 (1989).
2. Lecomte P, Dawes C. The influence of salivary flow rate on diffusion of potassium chloride from artificial plaque at different sites in the mouth. *Journal of dental research* **66**, 1614-1618 (1987).
3. Englander HR. The effects of saliva on the pH and lactate concentration in dental plaques. I. Caries-rampant individuals. *Journal of dental research* **38**, 848-853 (1959).
4. Abelson DC, Mandel ID. The effect of saliva on plaque pH in vivo. *Journal of dental research* **60**, 1634-1638 (1981).
5. Brown LR, Dreizen S, Handler S, Johnston DA. Effect of radiation-induced xerostomia on human oral microflora. *Journal of dental research* **54**, 740-750 (1975).
6. Almstahl A, Wikstrom M. Oral microflora in subjects with reduced salivary secretion. *Journal of dental research* **78**, 1410-1416 (1999).
7. Eliasson L, Carlen A, Almstahl A, Wikstrom M, Lingstrom P. Dental plaque pH and microorganisms during hyposalivation. *Journal of dental research* **85**, 334-338 (2006).
8. Leung KC, Leung WK, McMillan AS. Supra-gingival microbiota in Sjogren's syndrome. *Clinical oral investigations* **11**, 415-423 (2007).